# Improving the Health of the Homeless and How to Achieve It within the New NHS Architecture

**DOI:** 10.3390/ijerph17114100

**Published:** 2020-06-08

**Authors:** Paul Batchelor, James Kingsland

**Affiliations:** School of Medicine, University of Central Lancashire, Preston PR1 2HE, UK

**Keywords:** homeless, health improvement, collaboration, health determinants, organisation, primary care networks, care delivery

## Abstract

The publication of the National Health Service (NHS) Long Term Plan sees the creation of Primary Care Networks as the most appropriate solution to help improve overall health and address health inequalities. A key segment of society that suffers from poor health is the homeless. While the potential for the group to benefit from the NHS reform policy programme in England exists, it requires stronger collaborative working between the health and social care sectors Not least the new arrangements provide opportunities to tackle existing disease as well as the determinants of future ill health. However, if the policy vision is to be achieved, relations between the two sectors must occur and cross sector boundaries be broken down.

## 1. Introduction

In January 2019, NHS England published the NHS Long Term Plan [1]. The document set out proposals that cover the next 10-year period. A key focus of the work lies with ensuring the success of what are termed Integrated Care Systems (ICSs) and ‘new models of care’: the former addressing working relations between NHS organisations and local councils, the latter, termed Primary Care Networks (PCNs). The line of reasoning for the structural changes was that such bodies provided a better understanding of local needs and the most sensible approach to meet these was to “dissolve the historic divide between primary and community health services”. To help achieve this vision, primary medical and community services would receive a funding uplift of GBP 4.5 billion, which was to be invested by 2023.

PCNs would develop multidisciplinary teams working together to provide an extended range of services in a community setting, for a registered population of between 30,000–50,000 people. The arguments made for the creation of PCNs were many and included three key challenges: workforce shortages, poor integration with the wider care system and the need to improve ‘population health’.

Historically, care arrangements between the health and social care systems for the homeless have been divided. The legal framework for care is defined in the Homelessness Reduction Act (2017) that outlines the duty of care with respect to advice and information about homelessness that a local authority, namely the local councils in England, must provide [2]. Indeed, in paragraph 1.6 of the document, the wording stresses the need for joint planning and operational co-operation between social services authorities, health authorities and other agencies.

The gestation of PCNs as a structured unit has been lengthy, with one of the main proponents being the National Association of Primary Care (NAPC). Their guiding work on Primary Care Homes (PCH), a rapid test site approach and early evaluation process [3] helped to articulate the case for the subsequent implementation of PCNs. Prior to the adoption of PCNs, as the structural unit for the provision of care, the arrangements had used independent medical practices as the model through which care was provided. These varied enormously in size and in the level of engagement with the other sectors. The creation of Primary Care Networks (PCNs) sees the linking of General Medical Practitioners (GMPs) practices within a geographical area with community-based teams, using a designated single fund through which all resources flow. They embody a range of staff including GMPs, pharmacists, district nurses, community geriatricians and social workers.

The NAPC’s work was underpinned by four key principles: a combined focus on personalisation of care with improvements in population health outcomes; an integrated workforce, with a strong focus on partnerships spanning primary, secondary and social care; aligned clinical and financial drivers through a unified, capitated budget with appropriate shared risks and rewards, and; provision of care to a defined, registered population of between 30,000–50,000. The PCH population health management approach has three levels of personalisation: the wider community; a group of people with similar needs, and the person.

While the term ‘population health’ is currently used extensively in health policy and planning, an exact meaning has proved more elusive, yet is critical to establish the success or otherwise of policy goals. Murray et al. [4], in a review of measures of population health, suggested that the focus of work on measures should satisfy as many basic criteria and desirable properties as possible and argued that at minimum, five criteria should be used. Most importantly, they stated that:
“*Improving the estimation of summary measures of population health depends on designing the most appropriate measures for particular purposes*”.

This issue was restated by McCartney et al. [5], who in addition, made the important distinction between health and health inequalities, with the latter being defined as:
“*systematic, avoidable and unfair differences in health outcomes that can be observed between populations, between social groups within the same population or as a gradient across a population ranked by social position*”.

The World Health Organisation (WHO) has identified that while aggregated health measures have improved, health inequities remain [6]. In their report, the need for research in five areas was identified, four of which are applicable to the present agenda in England. The areas are:(a)How do societal and political structures and relationships differentially affect people’s chances of being healthy within a given society?(b)What are the interrelationships between factors at the individual level and within the social context that increase or decrease the likelihood of achieving and maintaining good health?(c)What are the characteristics of the health care system that influence health equity?(d)What were effective policy interventions to reduce health inequity in the above areas?

While the primary healthcare sector has had to address some of the above issues, the current focus of the sector remains based on finding medical solutions. Wright and Tomkins [7], in a review asking how the health needs of the homeless could be addressed, argued that while accessible and available primary health care is a pre-requisite for effective interventions, there is a requirement to overcome other barriers, not least the recognition that multi-agency working was necessary to enable homeless people to access the full range of health and social care services. This finding was also supported more recently by Hwang and Burns [8].

This paper builds on their work in the context of the new NHS architecture in England. While estimates of the number and categorisation of the homeless in England vary, the national membership charity working with people who have become homeless give the figure as over 250,000. These include rough sleepers, statutory and the ‘hidden’ homeless [9]. The present work aims to provide a framework for care planners and policy makers to help one of the most marginalised and hence disadvantaged groups, the homeless. To do this, this paper is divided into three further sections. First, an overview of the issues that the homeless face is presented. Second, the challenges of the present care arrangements in meeting the arising needs are discussed and finally, how the creation of PCNs provides opportunities to address the present shortcomings.

## 2. What Are the Problems for the Homeless?

A key focus of management within NHS England to date has centred on ensuring access to primary medical care for those in need. Shortcomings in achieving this goal arise for at least two reasons. First, the lack of a common framework for the user. Since the inception of the NHS, primary care provision has been delivered using four independent contractor services: general medical practice, community pharmacy, dental and optical services. For each of these services the user needs to meet certain requirements, for example, a defined address for registration. However, there is no consistency in the requirements by contractor services. This creates unnecessary and unhelpful barriers. Indeed, to help ensure successful outcomes of care, collaboration between the contractor sectors is required. Each sector cannot manage unaided the complex care required by people who do not have shelter. Second, a more general failing of the historical structure for health care delivery in the UK lies in its failure to tackle the wider determinants of health. The conditions that determine the health and wellbeing of individuals have historically lain outside of the conventional care system.

Fazel et al., in a review of the health of homeless people in high-income countries, reported that:
“*homeless people have higher rates of premature mortality than the rest of the population, especially from suicide and unintentional injuries, and an increased prevalence of a range of infectious diseases, mental disorders, and substance misuse. High rates of non-communicable diseases have also been described with evidence of accelerated ageing. Although engagement with health services and adherence to treatments is often compromised, homeless people typically attend the emergency department more often than non-homeless people*”.[10]

They went on to highlight that the most important structural determinant for homelessness was the lack of availability of low-cost housing. They suggested one approach to overcome this was to invest in programmes that focused on risk groups, including those leaving prisons or psychiatric hospitals. Such a proposal reinforces the need for joint working across the health and social sector.

In one of the few studies exploring visual health, Noel et al. [11] found that Canadian homeless youth had a high prevalence of visual impairment. This was despite being within a system of universal health insurance. They recommended that ongoing vision-screening programs, readily accessible free eye clinics, and particularly low-cost glasses could address this need, though they did caution extrapolation of the findings to other cities. The same group also found that the homeless had a high prevalence of hearing impairments [12]. Most importantly, they noted that hearing impairments have been shown to be strongly correlated with increased social isolation, reduced earning potential and higher rates of neurocognitive disease. These are key social determinants of health and are issues which must be addressed if the problem of homelessness is to be overcome.

The situation for oral health is remarkably similar. Daly et al. [13,14] in their work highlighted the poor levels of oral health in the homeless, with nearly 100% of the populations explored requiring dental care. Furthermore, a key finding of their work was to demonstrate how quality of life measures were far more appropriate than conventional clinical measures, for example, levels of tooth decay, in assessing health inequalities.

Their work also raised a fundamental issue when designing potential care arrangements. In the present NHS dental contract between the State and the care provider, one of the performance indicators concerns completed courses of treatment that the care provider managed. The homeless, when seeking care, prioritised pain relief and appearance as outcomes. If these were achieved to a level which the homeless found satisfactory, any remaining care on the course of treatment was seen as superfluous and the patient did not return to complete the prescribed care plan. Those responsible for managing holders of state dental contracts equated a high ratio of uncompleted to completed courses of treatment as poor performance. Care providers were subsequently penalised when compared to other contractor holders who chose not to provide care for the homeless. This highlights the importance of ensuring that all parties understand the complexities of care provision arrangements, not least the myriad of perspectives when assessing outcomes of care. Joint working between the sectors, enabling the issues of the care arrangements to be understood by all contractor groups, would help solve these problems.

## 3. Addressing the Problem

If the health needs of the homeless are to be adequately addressed, the solution must be more extensive than attempting to address their presenting treatment need. While such services can provide a solution to current clinical needs, unless the wider determinants of ill health are addressed, the problems will simply persist or reoccur. To address the wider determinants requires interventions that need collaboration between the current contractor groups. Whiteside [15] highlighted that without collaboration between social care and public health, the social and health inequalities would not be addressed. She went on to suggest that requirements to overcome this included a shared political analysis, a common language, and a framework for action.

There is considerable evidence to support the argument that when agencies work together, the resulting collaborative approach produces improved outcomes. D’Amour et al. [16], exploring frameworks used for collaborative working in the health care sector, identified five key concepts required: sharing, partnership, power, interdependency, and process. However, they also noted that their review of the literature did not establish how this could best be achieved for patients and this remains the challenge for the partners operating within the new PCN structures.

The need to address issues affecting collaboration was also identified by Gaboardi et al. [17]. Their work compared the goals of, and principles used by providers working with individuals who had experienced homelessness. Using qualitative methods in eight countries across Europe, they identified six common goals. These were:(a)support, i.e., helping people achieve their goals(b)integration, making people feel part of society(c)a requirement to meet basic needs such as food, and washing facilities(d)accommodation(e)well-being, both physical and psychological(f)jobs or activities during the day.

Stafford and Wood [18] have argued that factors which increased the prevalence of health problems for vulnerable and isolated individuals can be grouped into individual and social categories. For example, social factors can increase the probability that certain categories of the population will have poorer health and engage in unhealthy behaviours, including: lower socioeconomic status; lower education levels; lower resources and incomes; lack of social and family support; living in an unhealthy environment; and, limited access to preventive health care. Their work highlights again the need for cross collaboration between not only differing primary care contractor groups but with those working in social care, a key goal of the proposed PCN working arrangements.

If the current reform agenda for the NHS does not address health inequalities through improved action on the wider determinants of health, then the ambition of the NHS Long Term Plan is unlikely to be realised. Reducing health inequalities cannot be delivered through a traditional model focused primarily on healthcare provision, as has been highlighted as part of the unfinished business of the now completed NHS Five Year Forward View [19].

One of the most extensive reviews to identify what could be termed as best practice for improving health and addressing inequalities was the work done by the Expert panel on Effective Ways of Investing in Health (EXPH) [20]. In their report, they recommended that there was a need to focus away from addressing illness to assessing health and that intersectoral action and partnership working should be an integral aspect of achieving this. Not least, there was a requirement to support social mobilisation strategies, i.e., engage with the very groups any arrangements were designed to help, to ensure their views were heard.

Any successful approach, whilst having overall common principles, needs to adapt to the environment in which it is based. The care requirements of homeless people and solutions in one PCN area may not be the same as in another. The homeless may have variation in their demography or the rationale for finding themselves in their present position. As such, the determinants affecting health and well-being may be very different. Two key principles should therefore be used to underpin the work of PCNs in addressing the health needs and include:(a)Implementing models with proven transferability across different countries: the Housing First model has been adopted in a wide range of countries across the world, with the main aim of providing permanent housing to homeless people with high support needs [21].(b)Outreach programmes targeting specific health needs: the Find & Treat programme in the UK aims to locate and ensure treatment of tuberculosis among the socially vulnerable through a range of activities, including condition awareness raising, recruitment and training of peer advocates, treatment of Tuberculosis and provision of accommodation advice [22].

More specifically for primary care, Davies and Wood [23] recommended the need for several key structural elements in their work on addressing the health needs of the homeless in Australia. Among the elements were stable housing, continuity of health care, and outreach services. The new PCNs can potentially provide all of these, but only in conjunction with a local authority and, in particular, through the involvement of the social care sector.

## 4. Conclusions

The proposals for the NHS in England have seen the creation of Primary Care Networks. These structures theoretically provide improved opportunities to help ensure that the care needs of the homeless can be met. However, there is a need to overcome the structural and operational barriers that currently exist. Most importantly, the challenges of meeting existing health needs and addressing the social determinants of health remain. There will also need to be joint working between neighbouring PCNs as the boundaries of a statutory local council will not be coterminous. The population size served by an individual PCN is far smaller than that of a local council, yet the PCN will need to facilitate the joint working arrangements between health and social care required to deliver improved care for the homeless it provides care to.

If the opportunities for improvement of care for homeless people are to be realised, then a number of issues will need to be addressed. First, PCNs must recognise that to improve care outcomes for the homeless, the work must address the social and societal issues as well as their physical and mental health needs. Simply attempting to address clinical need, without altering the range of circumstances which gave rise to them, will not provide long term and sustainable improvements. A collaborative cross-care approach that sees the wider determinants addressed and provides a supportive environment to improve the socio-economic circumstances must form the focus of interventions. Second, there is a need to recognise that success of care may need to be measured at least initially, in a different way. Clinical measures can provide an indication of improvements in health status, but it is the qualities of life that are of equal importance. Providing care to improve the confidence and wellbeing of the individual, enabling them to seek employment or form relationships that improve their integration into wider society are critical. Third, as with all care arrangements, there is a continual need to provide education and training for staff across disciplines to ensure that the opportunities provided by the reforms are realized in an efficient and effective manner.

Such proposals are consistent with the recommendations of Briggs et al. [24]. They concluded, when exploring the role of integrated care systems and sustainability and transformation plans identified in the NHS Long Term Plan, that care delivery organisations are to “*…prevent disease and improve population health, they need to look beyond their 2016 plans and fill the gaps in the Long Term Plan on social determinants*”.

The issues of homelessness are a societal problem that the care system has historically struggled to overcome. The newly planned restructuring of the architecture of the NHS provides an opportunity to address previous shortcomings which local agencies must seize; the professions can and must cross previous organisational boundaries to work towards doing so. However, as Vrijhoef summarised in a recent editorial:
“*First, care coordination should no longer be offered as a one size fits all programme to those in need. Second, care coordination for patients with very high use of health care services should go beyond health care to include other critical needs, including housing and legal support. Third, offering care coordination without fixing the system that causes fragmentation from the start is unsustainable*”.[25]

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
