# Peer review of "Improving the Health of the Homeless and How to Achieve It within the New NHS Architecture"

_ijerph, 2020, doi:10.3390/ijerph17114100_

Round 1

Reviewer 1 Report

This paper contains many grammatical and punctuation errors, and much of what the authors are trying to convey is not clear. The material seems interesting, but the thought processes and the direction of the paper are not easy to follow.

Comments for Improving the Health of the Homeless and How to Achieve it within the New NHS Architecture.

Abstract: I would rewrite much of the abstract. For the first sentence, maybe use a different word than “opportunities.”

The second sentence is not grammatically correct.

I would revise the second sentence to state: “This paper argues that if health improvements are to occur, then both the care and the determinants (of _____) must be addressed, and this requires a collaborative approach that the creation of Primary Care Networks provide.

  1. Introduction:

Paragraph 1; Line 5: add comma: “England, which”

Paragraph 2; Line 2-3: should there be a comma after Homes and before a rapid test; and after process?

Paragraph 2; Line 10: use a ; after “needs,” and remove ; before “the person.”

Paragraph 3: The first sentence does not make sense. I’m not sure what it’s saying.

Paragraph 3; Line 3: add comma after “Murray et al.” and after “population health.”

Paragraph 4: Line 2: remove one of the periods form the sentence.

Paragraph 5: Line 1: add a comma after “et al.” and after “in addition”

Paragraph 6: Most of this paragraph does not make sense as worded.

Paragraph 7: Line 3-4: needs commas; and this sentence would read better if it were broken into multiple sentences.

Paragraph 8: last sentence: how what can be overcome?

  1. What are the problems for the homeless? (perhaps use a question mark here)

Paragraph 1: access to primary medical care for who? This fails as a solution to what?

Paragraph: Line 3: “Since its inception……” this sentence is too long and confusing- needs to be clearer and made into multiple sentences.

Fazel et al. (add comma) ……in high-income countries (add comma) ….

Paragraph 3: revise; it’s not clear and the grammar is incorrect.

Paragraph 4: Line 1: add comma after “health,”

               Line 4: add comma between “need” and “though”

               Line 5-6: “The same group…” this sentence needs revision.

Paragraph 5: Line 1: remarkedly similar to what? Sentence 2: remove “in their work”

               Sentence 3: revise sentence.

               The majority of this paragraph is not clear and needs revision.

Paragraph 6: Revise.

  1. Addressing the Problem

Paragraph 1: Line 1: revise

Sentence 3: unclear

Cite Whiteside appropriately

Paragraph 2: Sentence 3: revise/unclear

               Sentence 4: needs to be more specific

Paragraph 6: change “LTP are” to “LTP is”

Paragraph 7: add comma after “In their report”

  1. a) needs a period at the end.

Paragraph 11: add comma after “particular”

  1. Conclusions

Paragraph 1: Sentence 3 needs commas.

               Sentence 4: might be better if it is made into multiple sentences.

Paragraph 2: Sentence 1 is not a complete sentence.

               Line 8: revise sentence…… “health status, the qualities of life are of equal importance.”

               Line 9: replace comma with “and”

               Line 10: add a comma after “care arrangements”

Paragraph 3: Sentence 1 needs commas and should be revised into multiple sentences.

Paragraph: this sentence has multiple errors.

The final sentence of the paper is not clear.

Also, there should be a period after “This research received no external funding.”

Author Response

We would like to both apologise and thank the referees for their comments regarding the original submission. The initial submission was below standard and it is only through the kindness of the referees in their work that the opportunity to resubmit has arisen.

First we would like to address the general comments made. We have corrected and improved the language, punctuation and grammar throughout the document. This hopefully has addressed the valid concerns of both referees, not least providing the potential readers with clarity in the points we are trying to make.

More specifically we have made a number of changes to address concerns raised.

In the introduction we have undertaken a major rewrite that provides a better understanding of the reforms in the NHS in England including a comment on the legal framework for care provision of the homeless.

We have revised the second section (What are the problems for the homeless?) outlining some of the aspects which influence care delivery. We have not explored in detail the PCNs as while we understand the question raised by one of the referees concerning their functioning, we wish to examine the potential opportunities that exist given it is Government policy. The issue of analysing the abilities of PCNs to undertake the tasks is very relevant but we feel it is for a separate paper.

We have made a number of other changes in subsequent sections that we think add clarity to the issues we would wish to make.

We thank the referee for the suggestion of adding something concerning the scale of homelessness and have provided a brief description to help frame our work.

Finally, we have altered the abstract to help address the concerns raised.

Reviewer 2 Report

The article deals with an important topic such as health inequalities and homeless people's access to healthcare.

The Authors analyze a case study i.e. the National Health Service in England and its recent re-arrangements with a particular focus on the creation of Primary Care Networks (PCNs).

However, in my view, the paper present several severe limitations that should be addressed and fixed before publication:

  • the first one is that the paper does not help the reader to understand how the NHS in England works and what are the recent re-arrangements. For example, except for the implementation of the PCNs, are there other changes occurring? In which direction? Access, availability of services, affordability...? 
  • The second one, maybe the most severe, is that the Authors did not critically analyze - actually they did not even describe - the PCNs that are supposed to be the focus of the article.
  • Finally, a brief overview on how many homeless people are present in England and their characteristics is missing.

Specific comments:

  • in the text there are several grammatical mistakes such as semicolons between "and" and the subsequent word (for example in the "Introduction" section, pp. 1-2);
  • p. 3 section "Addressing the problem": the Authors cite Whiteside but the year of the article/book is missing.

To conclude, the article presents serious flaws and needs a strong effort to improve it.

Author Response

(The authors gave the same response as above.)

Round 2

Reviewer 1 Report

I appreciate the revisions made, and I believe that the paper is publishable.

Reviewer 2 Report

The Authors did a significant effort to take into consideration Reviewers' comments.

I think the paper is now ready for publication.